# Wastewater Treatment and Wood Production of Willow System in Cold Climate

**Ganbaatar Khurelbaatar** [1],*, **Manfred van Afferden** [1], **Christopher M. Sullivan** [1], **Christoph Fühner** [1], **Jamsaran Amgalan** [2], **Jöerg Londong** [3] **and Roland Arno Müller** [1]

1 Centre for Environmental Biotechnology, Helmholtz Centre for Environmental Research—UFZ, 04318 Leipzig, Germany; manfred.afferden@ufz.de (M.v.A.); christopher.sullivan@ufz.de (C.M.S.); christoph.fuehner@ufz.de (C.F.); roland.mueller@ufz.de (R.A.M.)

2 Department of Technology, Mongolian University of Science and Technology, Ulaanbaatar 14191, Mongolia; oij_amgaa@yahoo.com

3 Chair of Urban Water Management and Sanitation, Bauhaus Universität Weimar, 99421 Weimar, Germany; joerg.londong@uni-weimar.de

* Correspondence: ganbaatar.khurelbaatar@ufz.de; Tel.: +49-341-235-1842

**Abstract:** This article studied how wastewater treatment performance of a short rotation forestry system was influenced by the seasonal operational changes under the extreme Mongolian winter conditions. For this reason, two beds planted with Willow (Salix *spec*.) and Poplar (Populus *spec*.) trees were operated over a period of two years under two different seasonal conditions: (A) "external winter storage" and (B) "internal winter storage" of pretreated wastewater. For operational condition A, the tree-bed was loaded with wastewater for only 4 summer months. For this operational condition it was considered that the treatment bed was fed with primary treated wastewater, which was stored in a sealed pond during the remaining 8 months. The other Bed B was irrigated throughout the year (12 months) with the same daily loading rate. In winter, the wastewater accumulated as ice in the tree-bed. Bed A, with external winter storage, showed mass removal percentage up to 95%, while the bed with internal winter storage showed mass removal rates up to 86% for pollutants such as COD, BOD$_5$, TN, and TP. A high yield of biomass was recorded for both beds with slight differences. Based on the results, a design recommendation was developed for full-scale systems of short rotation coppice irrigated with wastewater under various operational conditions, which show these systems to be a viable method for treating wastewater and producing biomass for energy production in Mongolia.

**Keywords:** wastewater treatment; wastewater reuse; wood production; land application; short rotation coppice; and cold climate

## 1. Introduction

Extreme cold and long winters provide a significant challenge to wastewater management in Mongolia. While a number of wastewater technologies have been tested in the region, there is a pressing need for simple and cost-efficient treatment alternatives that might offer reliable performance and allow resource recovery under such conditions. An application of primary treated wastewater on short rotation forestry system has been tested and shown to be an attractive option for both wastewater treatment and biomass production [1]. While studies showed generally a high efficiency of such systems under cold climate conditions [1,2], the exact influence of the extreme winter conditions on operational requirements have not been well studied.

Land application of wastewater is recognized as being a robust, simple, and reliable technology for wastewater treatment in many parts of the world. This approach has been successfully combined with the short rotation coppice of Willow, resulting in enhanced treatment performance and beneficial biomass production under many different climatic

conditions [3]. However, a concern with land application technology has been raised during the winter period in relation to operation and performance. Low temperatures and ice formation result in deterioration of both physical structure and biological performance of the system due to an inhibition of microbial activity in the soil [4]. In addition, the biomass yield might be negatively affected by the extended period of ice formation. Only a few operational options have been reported for land application of wastewater during winter.

The first option is to operate the system only during summer months. This implies that the wastewater is either stored or treated in a conventional wastewater treatment plant during the winter period [5]. The city of Enköping in Sweden, with a population of 20,000 inhabitants, uses 80 ha of tree plantation as part of its conventional treatment approach for domestic wastewater. Timber generated from the site is used in a local district heating system. The tree plantation is irrigated in summer with treated wastewater only. In winter, the treated effluent is discharged to a river [6]. In Huolinguole city, China, primary treated wastewater is used for summer-irrigation of different tree species including Larix, Pinus and Populus over an area of 880 ha. In wintertime, the flow of 10,000 m$^3$ per day of municipal wastewater is stored in a reservoir (external winter storage) [7] (Figure 1). This option might be suitable for wastewater treatment and tree growth under the climatic conditions in Mongolia. However, "external winter storage" (EWS) implies additional costs for the construction of sealed reservoirs.

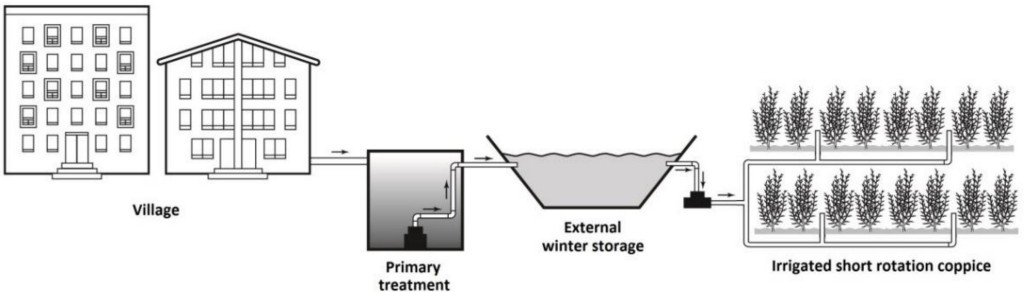

**Figure 1.** Option A: Operation of a wastewater irrigated short rotation coppice system with "external winter storage".

The second option is continuous operation throughout the year. The US EPA [4] identifies several successful operations of land treatment systems in cold tempered areas in the Northern U.S. and their continuous year-around operation under appropriate maintenance. In this case, higher hydraulic loading is applied to maintain the temperatures within the system to avoid freezing of the soil layer. Ice cover and snow serve as an additional insulation, while the soil temperature is maintained around 0 °C. In US-EPA [8] it is noted that soil infiltration may occur even when the soil temperature drops to minus 4 °C. Forest crops have been identified as more suitable for this approach compared to agricultural crops. However, a land treatment system in Central Vermont showed significantly lower treatment performance in winter with a three times higher concentration of BOD, TSS, and TN in drainage water compared to summer operation [4]. This option however might not be suitable for the cold and long Mongolian winters due to the fact that the winter temperatures in Mongolia are substantially lower than those of Vermont and other reported cold temperate regions.

The third option would be the use of "internal winter storage" of the wastewater. This approach is similar to the Zero Discharge Systems reported by US-EPA [4], which receives the same hydraulic load of primary treated wastewater throughout the year, whereas on the Willow-plantation, the wastewater is stored in wintertime as ice cover which is dyked and sealed using a subsurface base liner (Figure 2). This treatment technology is mainly applied as an on-site system for individual houses. For treatment systems on a larger scale, such as villages or small towns, a subsurface base liner (e.g., HDPE-geomembrane) would economically not be feasible due to the high area requirements. For this reason, we

have proposed internal winter storage of primary treated wastewater on the dyked tree plantation without a base liner. This option might be an attractive solution for Mongolia due to the fact that it does not require the costly external winter storage.

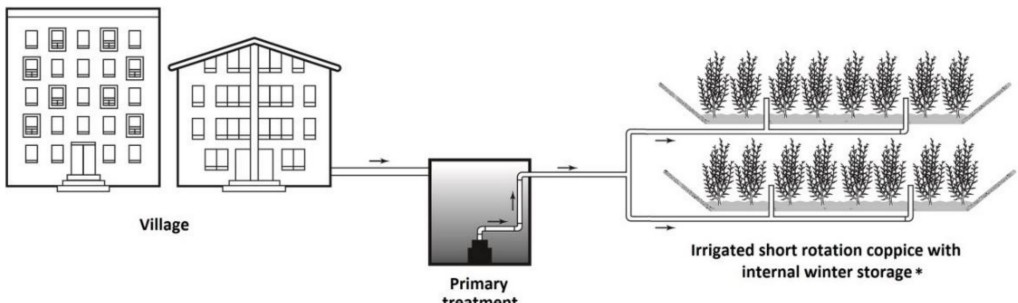

**Figure 2.** Option B: Operation of a wastewater irrigated short rotation coppice system with "internal winter storage" * The proposed internal winter storage is not sealed with a base liner but dyked along the perimeter.

The objective of this study has been to investigate the treatment performance of short rotation coppice systems of Willow and Poplar for wastewater treatment under two different seasonal conditions: (A) external winter storage, and (B) internal winter storage as presented in Figures 1 and 2.

## 2. Materials and Methods

### 2.1. Site Description

The pilot plant for short rotation coppice of Willow (*Salix spec.*) and Poplar (*Populus Spec.*) irrigated with primary treated wastewater was established in northern Mongolia (49°27′31.95″ N, 105°58′42.92″ E) in the summer of 2011 and tested for two consecutive years. The climate conditions of the site can be characterized as having cold winters and short hot summers with mean daily temperatures ranging from −33 to −15 °C in January and from 10 to 32 °C in July, respectively. Additionally, the low precipitation (with a long term mean amount of 260 mm) and a low mean annual temperature (−1.5 °C) are the main characteristics of the climate conditions of Mongolia. The climate and precipitation data throughout two years are described in Khurelbaatar [1].

### 2.2. Pilot Plant

The pilot plant consisted of primary treatment, tree planted irrigation beds that are sealed with a base liner and equipped with drainage pipework, and a sampling manhole equipped with tipping counters. Despite the idea of tree planted areas irrigated with wastewater not sealed with a base liner, the sealing was necessary for the pilot beds in order to catch the treated effluent (drainage water) for quantification and chemical analysis.

Municipal wastewater was treated at the pre-treatment tank (8 m$^2$) and pumped to the beds. Both beds had an effective depth of a 1 m local sandy loam soil and a spare space of 1 m on top for winter ice accumulation. The surface area of the beds was 15.75 m$^2$. The beds were equipped with a simple surface irrigation pipe (25 mm), and drainage pipework (100 mm) at the bottom of the soil layer. To enable the ice accumulation on the beds, the irrigation pipe stood out approximately 1.1 m over the soil surface (Figure 3). The beds were fed with primary treated wastewater via a pump controlled by a timer with a load of 15 mm·day$^{-1}$. One bed (A) was loaded only during summertime (from mid-May through mid-September), thus representing option A "external winter storage", while the other bed (B) was irrigated throughout the year and thus represented option B "internal winter storage". The bed B with internal winter storage accumulated ice during winter, which was followed by a peak in drainage water caused by ice melt (Figure 4).

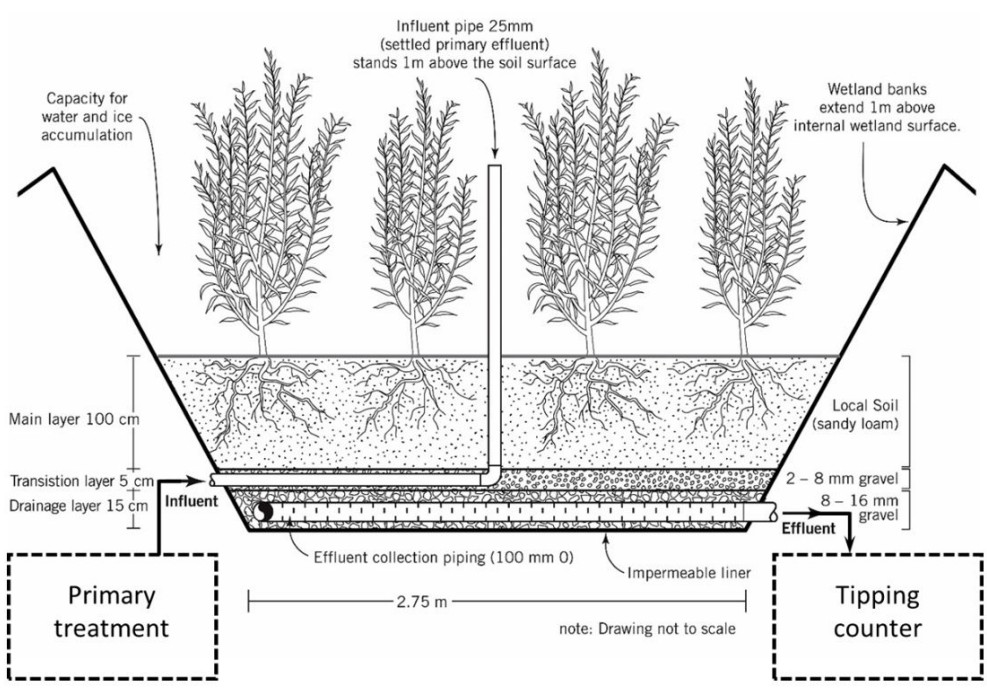

**Figure 3.** The schematic of the pilot plant and the cross sectional view of the tree planted beds.

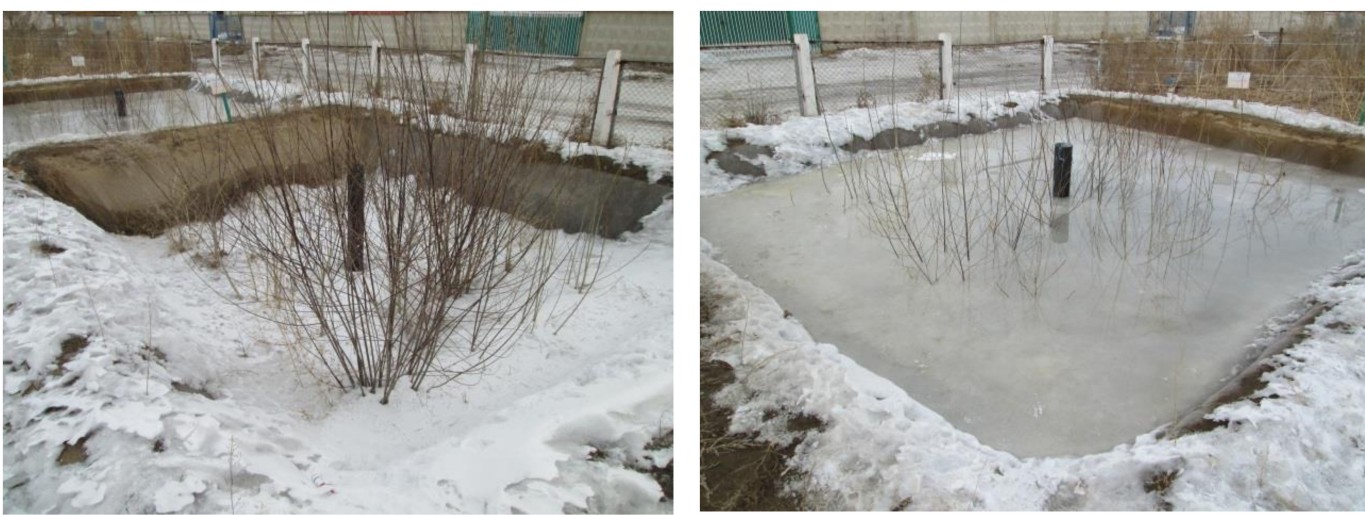

**Figure 4.** The pilot treatment beds. Bed A operated with external winter storage and summer irrigation (left) and bed B operated with internal winter storage, where the wastewater is stored on top of the treatment area during winter.

### 2.3. Analytical Methods

Water samples were taken weekly from the effluent of the beds and the primary settling tank. The pH was measured using a Multi 340i with a SenTix 41 sensor on site, while EC and DO were measured using the same equipment with a Tetracon 925 sensor. The remaining parameters, except *E. coli*, were measured in the water lab located next to the pilot plant. COD, TN, $NH_4$-N, $NO_2$-N, $NO_3$-N, $PO_4$-P, TP were measured using Hach-Lange test kits (Hach-Lange GmbH, Düsseldorf, Germany), thermostat LT200 (Hach-Lange GmbH, Düsseldorf, Germany) and spectrophotometer DR 2800 (Hach-Lange GmbH, Düsseldorf, Germany). $BOD_5$ was determined by using OxiTop IS-6 (WTW GmbH, Weilheim, Germany) units in accordance with the equipment manual and the German standard DIN EN 1899-2 [9]. The *E. Coli* analysis was carried out in accordance with the

Mongolian standard MNS 5668 [10]. The water quality parameters and the analytical methods are summarized in Table 1.

**Table 1.** Overview of water quality parameters and analytical methods.

| Parameter | Unit | Method | Equipment | Remark |
|---|---|---|---|---|
| pH | - | Standard method | SenTix 41/Multi 340i | |
| EC | $\mu S \cdot cm^{-1}$ | Standard method | TetraCon 925/Multi 340i | |
| DO | $mg \cdot L^{-1}$ | Standard method | TetraCon 925/Multi 340i | |
| COD | $mg \cdot L^{-1}$ | LCK 314, 15–150 $mg \cdot L^{-1}$ | LT200 and DR 2800 | |
| TN | $mg \cdot L^{-1}$ | Laton LCK 238, 5–40 $mg \cdot L^{-1}$ | LT200 and DR 2800 | |
| $NH_4$-N | $mg \cdot L^{-1}$ | LCK 303, 2–47 $mg \cdot L^{-1}$ | DR 2800 | Filtered |
| $NO_2$-N | $mg \cdot L^{-1}$ | LCK 341, 0.015–0.6 $mg \cdot L^{-1}$ | DR 2800 | Filtered |
| $NO_3$-N | $mg \cdot L^{-1}$ | LCK 339, 0.23–13.5 $mg \cdot L^{-1}$ | DR 2800 | Filtered |
| $PO_4$-P | $mg \cdot L^{-1}$ | LCK 349, 0.05–1.5 $mg \cdot L^{-1}$ | DR 2800 | Filtered |
| TP | $mg \cdot L^{-1}$ | LCK 349, 0.05–1.5 $mg \cdot L^{-1}$ | LT200 and DR 2800 | |
| $BOD_5$ | $mg \cdot L^{-1}$ | DIN EN 1899-2 (1998) | OXiTop IS 6 | |
| *E. coli* | $MPN \cdot (100\ mL)^{-1}$ | MNS 5668: 2006 | | |

The composition of pretreated wastewater used for irrigation was as follows: pH: $7.6 \pm 0.3$, EC: $0.81 \pm 015\ dS \cdot m^{-1}$, COD: $152 \pm 57\ mg \cdot L^{-1}$, $BOD_5$: $104 \pm 27\ mg \cdot L^{-1}$, TN: $56 \pm 19\ mg \cdot L^{-1}$, $NH_4$-N: $45 \pm 16\ mg \cdot L^{-1}$, TP: $4.6 \pm 1.7\ mg \cdot L^{-1}$, and *E. coli*: $1.67E + 07 \pm 1.62E + 07\ MPN \cdot 100^{-1} \cdot mL^{-1}$.

The tree growth, biomass yield, and the nutrient content of the biomass were measured on an annual basis. The detailed procedures of water and biomass analysis were reported by Khurelbaatar [1].

Water loss due to evapotranspiration was calculated using Equation (1):

$$\text{Water loss} = [(\text{precipitation} + \text{irrigation water}) - \text{drainage water}] \tag{1}$$

Weekly irrigation water volume, drainage water volume and water loss due to evaporation and/or evapotranspiration were summed up for the estimation of an annual cumulative water balance. Here it is important to point out that precipitation was calculated by using a factor of 1.25 due to the difference in the precipitation receiving area and the treatment area. Due to the banks of the beds extending 1 m above the planted soil filter, there was no occurrence surface runoff and thus ignored in the equation.

The specific mass of the wastewater compounds (CSB, $BOD_5$, TN, $NH_4$-N, $NO_3$-N and TP) along with irrigation and drainage water was calculated using the respective concentration ($mg \cdot L^{-1}$) of each compound multiplied by the volume (L) of irrigation and drainage water on a weekly basis. The mass removal rate for the compounds was calculated as the difference between the inflow mass (irrigation water) and the outflow mass (drainage water). The weekly mass removal rates were summed up for calculating the cumulative annual or monthly mass removal rates.

The sampling and analysis were carried out throughout two consecutive years between June 2012 and May 2014, where water analysis was carried out on a weekly basis and the biomass yield was measured at the end of each growing season. The mass balance results of the two study years were presented in this paper in order to evaluate the treatment performance of the system.

## 3. Results

### 3.1. Water Balance

Figure 5 shows the water balance of the two beds on an annual basis, while the monthly water balance is presented in Figure 6. Bed A (external winter storage) was designed to be fed with primary treated wastewater only during the summer period (mid-May to mid-September), while bed B (internal winter storage) was loaded throughout the year. The mean annual irrigation loads were as high as 2020 and 3770 mm for bed A and B,

respectively. When including the precipitation, the mean annual water loads were as high as 2560 and 4300 mm (Figure 5) for bed A and bed B, respectively.

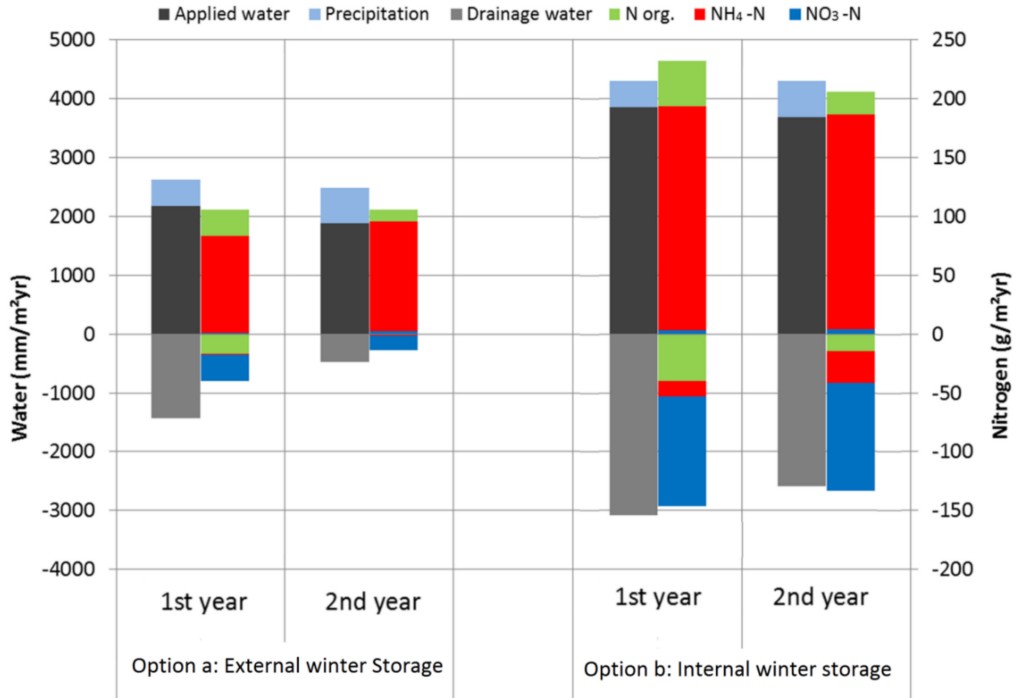

**Figure 5.** Annual water balance and nitrogen composition of the application and drainage waters within the beds.

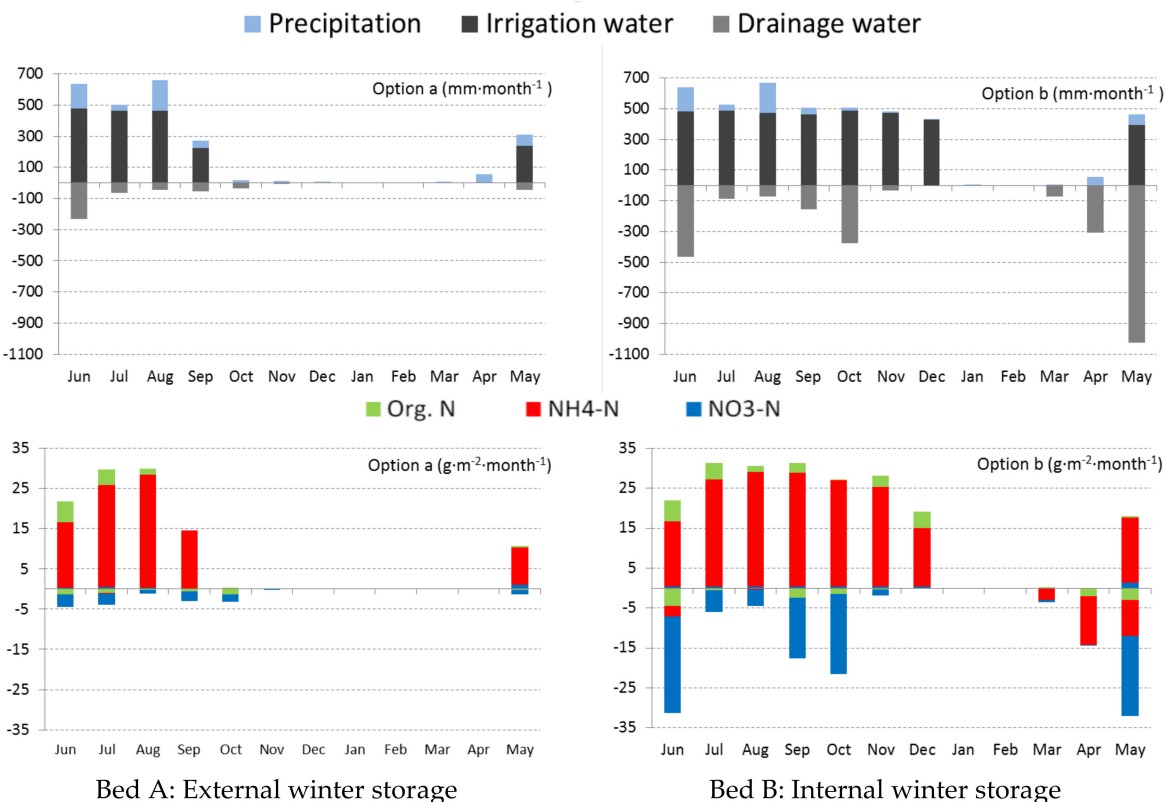

**Figure 6.** Monthly water and nitrogen balance investigated within the second year. The TN composition is presented in terms of Org. N, $NH_4^+$-N and $NO_3^-$-N (Note: The positive values denote the irrigation water and applied nitrogen and the negative values denote the drainage water and the leached nitrogen).

In the case of bed B, intended for continuous irrigation throughout the year, irrigation was reduced between January and March due to an unexpected long winter and insufficient ice storage capacity in the bed. During the winter period (October to March), no identifiable drainage occurred and irrigation water and precipitation accumulated on top of the beds as an ice cover. In spring (April to May), during ice melt, a peak period of drainage water was detected (Figure 6). In the second study year, the total amount of drainage water recorded during April and May was as high as 1400 mm, accounting for 56% of the total annual drainage water, which in turn accounted for 33% of the total hydraulic load. During the summer period (May to September), a similar pattern of drainage was observed for both beds (Figure 6). The amount of drainage water during this period amounted to 770 and 520 mm for bed A and B, respectively. However, bed B showed a high peak of drainage in October, which might have been caused by a decreased evapotranspiration rate and continuous loading during that time. The annual water loss due to evapotranspiration (ET) for the two treatment beds was calculated to be 1193 and 1135 mm for the first year and 2008 and 1800 mm for the second year, respectively.

*3.2. Mass Removal Rate*

3.2.1. COD and BOD$_5$

For the first year, the COD mass removal rate was determined to be 155 and 392 g·m$^{-2}$·y$^{-1}$ for bed A and B, respectively, resulting in a removal percentage of 60% and 68%, respectively. COD mass removal rates of 283 and 403 g·m$^{-2}$·y$^{-1}$ with respective removal percentages of 91% and 63% were found in the second year (Table 2). The bed B showed a mass removal rate higher than that of bed A, due to the higher COD load. The removed percentage was similar for both beds in the first year. In the second year, the mass removal rate for bed A increased drastically, whereas it remained stable for bed B. This sudden increase in the removal rate for bed A might be explained by the fact that almost no drainage water occurred during the second year.

**Table 2.** Cumulative mass loading and removal rate of the contaminants recorded during both the study years within the beds.

| Parameter | Study Year | Option A: External Winter Storage | | | Option B: Internal Winter Storage | | |
|---|---|---|---|---|---|---|---|
| | | Load (g·m$^{-2}$) | Removal (g·m$^{-2}$) | Removed Percentage (%) | Load (g·m$^{-2}$) | Removal (g·m$^{-2}$) | Removed Percentage (%) |
| COD | 1 | 258 | 155 | 60 | 573 | 392 | 68 |
| | 2 | 311 | 283 | 91 | 642 | 403 | 63 |
| BOD$_5$ | 1 | 128 | 118 | 92 | 212 | 186 | 88 |
| | 2 | 194 | 189 | 97 | 390 | 323 | 83 |
| TN | 1 | 105 | 65 | 62 | 232 | 86 | 37 |
| | 2 | 106 | 88 | 84 | 206 | 76 | 35 |
| NH$_4^+$-N | 1 | 82 | 82 | 99 | 190 | 177 | 93 |
| | 2 | 93 | 93 | 100 | 183 | 156 | 85 |
| TP | 1 | 8.6 | 3.2 | 37 | 19.1 | 7.8 | 41 |
| | 2 | 9.4 | 7.8 | 83 | 18.3 | 9.4 | 52 |

The BOD$_5$ removed percentages were 92% and 88% for the first year and 97% and 83% for the second year for bed A and B, respectively. In general, bed B, fed with a higher load, showed increased mass removal rates of COD and BOD$_5$. On the contrary, the removed percentage was found to be higher for bed A.

3.2.2. Nitrogen

Nitrogen data analysis revealed that the total nitrogen TN removal showed a very similar pattern for the two beds (Table 2 and Figure 4). The total nitrogen (TN) mass removal rate for the summer-operated bed A was found to be 65 g·m$^{-2}$·y$^{-1}$ in the first

study year and 88 g·m$^{-2}$·y$^{-1}$ for the second study year, which resulted in a TN mass removal percentage of 62% and 84%, respectively. The TN mass removal rate within Bed B, operated year-round, was as high as 86 and 76 g·m$^{-2}$·y$^{-1}$ resulting in a removal percentage of 37% and 35% during the first and the second study year, respectively (Figure 4 and Table 2).

In terms of ammonium nitrogen, both beds showed high rates of nitrification. The lowest NH$_4$-N removal was around 85% for bed B (Table 2). In this bed, 100% of the ammonium associated with drainage water was found during the ice melt period in springtime (April-June) (Figure 6). The nitrate nitrogen found in drainage water was much higher for bed B compared to bed A (Figure 5). This can be attributed to the ice melt and the high amount of drainage water from the bed with internal winter storage. The TN, NH$_4$-N, and NO$_3$-N peaks in April and May were as high as 38%, 90%, and 30% of the total amount in drainage water in the second year for the bed B, respectively.

### 3.2.3. Phosphorus

The TP mass removal rates for bed A were 3.2 and 7.8 g·m$^{-2}$·y$^{-1}$ (with 37% and 83% removed percentage) for the first and second year, respectively, and amounted for the bed B to 7.8 and 9.4 g·m$^{-2}$·y$^{-1}$ (with 41% and 52% removed percentage), respectively (Table 2).

A high peak of TP in drainage water also occurred during the ice melt period in bed B. Around 20–55% of the TP was found during the ice melt period for this bed in the drainage water (Figure 6). The TP removal rate increased slightly for bed B, while for bed A, it showed a sudden increase following the pattern of COD and TN removal. This might be explained again by the low amount of drainage water occurring in bed A in the second study year. In general, the increase of TP removal can be attributed to increased ET and additional tree biomass in the second year.

### 3.2.4. *E. coli*

While bed A showed an *E. coli* mass removal of 1.2–1.5 log reduction, bed B showed one log lower (0.4–0.7) removal rates (Table 3). The higher rate of *E. coli* removal for bed A might be explained by the fact that the bed A was only operated during summertime, which provides favorable conditions for *E. coli* removal in the upper soil layer due to the soil microbial activity at a favorable temperature [11] and prolonged hydraulic retention time [12].

**Table 3.** Cumulative *E. coli* in the irrigation water, drainage water and their removal recorded for both study years within the beds.

| Parameter | Study Year | Bed A: External Winter Storage | | | Bed B: Internal Winter Storage | | |
|---|---|---|---|---|---|---|---|
| | | Irrigation water | Drainage water | Log reduction | Irrigation water | Drainage water | Log reduction |
| *E.coli* | 1 | 1.26E + 11 * | 8.18E + 09 * | 1.2 | 2.02E + 11 * | 4.41E + 10 * | 0.7 |
| | 2 | 3.09E + 11 ** | 1.04E + 10 ** | 1.5 | 6.31E + 11 ** | 2.57E + 11 ** | 0.4 |

\* CFU·m$^{-2}$·a$^{-1}$; \*\* MPN·m$^{-2}$·a$^{-1}$.

### 3.3. Tree Growth

At the end of the second study year, both beds showed similar tree growth. Bed A showed a 100% survival rate for both Willow and Poplar trees, whereas bed B showed a 90% and 100% survival rate for Willow and Poplar trees, respectively, indicating that Willow trees might have some sensitivity to the accumulation of ice. A similar bed with internal winter storage and lower loading rate (5 mm·day$^{-1}$), which was tested during the same time, indicated a slightly lower survival of the trees [1]. This might suggest that the trees are more susceptible to dry conditions than the cold [13]. The mean height of the Willow trees reached 3 and 3.3 m for bed A and B, respectively. The mean height of Poplar also reached 3.4 m and 3 m in the respective beds. The same pattern could be observed for

the tree diameter. The Willow diameter was 17 and 15 mm for bed A and B, respectively, while the respective Poplar diameter reached 24 and 38 mm.

When the trees were cut and the biomass measured, bed B showed a yield of 19.6 kg DM for Willow (9 trees) and 21.2 kg DM for Poplar (10 trees). For bed A, the biomass yield for Willow and Poplar trees amounted to 37.4 kg DM (10 trees) and 16.0 kg DM (10 trees), respectively. The total biomass yield for bed A and B was 3.4 and 2.8 Kg/m$^2$ respectively.

## 4. Discussion

### 4.1. Treatment Performance of the Wastewater Irrigated Short Rotation Coppice System in a Cold Climate

The results demonstrate that a short rotation coppice system with Willow and Poplar can function as a wastewater treatment and wood production technology in cold climate regions under two different seasonal conditions. Whereas a number of studies have been carried out thus far under conditions similar to bed A with external winter storage, bed B with internal winter storage has distinct differences with respect to any other waste water irrigated short rotation coppice system reported in the literature.

Despite having long and cold winters, Mongolia also experiences short and hot summers, which has proven to lead to high ET in such systems [1]. For bed A, this high ET leads to minimal drainage water, which positively influences the treatment performance of the wastewater. A similar bed, fed throughout the year with a much smaller amount of pre-treated wastewater (5 mm·day$^{-1}$), showed an ET which was very similar to bed B operated throughout the year [1]. The slightly higher ET found in bed B compared to bed A might be explained by the fact that there was no ice accumulation followed by a peak drainage from this particular bed. The ET recorded during this study is consistent with that found by [14] in Scandinavia and [15] in Italy, proving that this wastewater treatment system can function with high efficiency even in colder temperate countries, such as Mongolia.

In terms of organic load, bed A demonstrated both high mass removal rate and removed percentage while bed B showed much higher mass removal rate but low removed percentage. In the US-EPA [3] it is stated that the main removal process for COD and BOD$_5$ in wastewater land treatment systems occurs on the soil surface or close to the soil surface layer and that the application rate of 500 kg BOD$_5$·ha$^{-1}$·d$^{-1}$ is not a limiting factor for a slow rate land treatment system. Paranychianakis et al. [16] reported an efficient removal rate of more than 90% at a BOD application rate of 330 kg·ha$^{-1}$·d$^{-1}$. In the present study, the maximum application rates of 18 kg COD·ha$^{-1}$·d$^{-1}$ and 11 kg BOD$_5$·ha$^{-1}$·d$^{-1}$ were much lower than the reported values in published data, suggesting that the treatment beds were not overloaded. On the contrary, the low removal percentage found in bed B, which was operated throughout the year, reflects the peak drainage and the contained mass of COD and BOD$_5$.

In the present study, the mean annual application rate with regard to TN and TP for bed A with external winter storage was as high as 1050 and 55 kg·ha$^{-1}$, respectively. Thereby, it is in the same range to that described for land treatment systems with a mean annual application rate of TN and TP of 1170 and 104 kg·ha$^{-1}$, respectively [17]. Bed B with internal winter storage, operated throughout the year, was fed with a much higher load of nutrients; 2190 and 187 kg·ha$^{-1}$ TN and TP, respectively. Tzanakakis et al. [18] and Holm and Heinsoo [19] reported higher removal efficiencies for TN and TP (95% and 84%, respectively) at this range of nutrient load, which might be explained by the fact that these studies were conducted in Greece and Estonia, which have a warmer year-long climate compared to Mongolia. Bed B, showing TN removal percentage of only 35–37%, might suggest that this particular bed may have been overloaded with nutrients. Another bed operated under same condition (of internal winter storage) with a lower loading rate (of 5 mm·day$^{-1}$) exhibited TN and TP removal percentage of 80% and 85%, respectively [1]. In general, both beds showed high mass removal rates, highlighting the removal capacity of such a system in a cold climate.

In general, land treatment systems are distinguished by a high treatment performance for E. coli [3]. The low treatment performance shown in the present study in the case of bed B might be explained by the unique character of the wastewater treatment system, which accumulates the wastewater during wintertime, followed by the ice melt effect. Tzanakakis et al. [11] reported in his study a reduction of up to 6 logs for E. coli, whereas this investigation (similar to other studies) was conducted under much warmer climatic conditions and at lower loading rates.

One of the benefits of the wastewater treatment technology described here is the simultaneous production of biomass, which is in high demand in Mongolia, considering the cold winters. Conventional short rotation forestry systems have a biomass yield of 6–8 t DM$\cdot$ha$^{-1}\cdot$y$^{-1}$ [20]. When irrigated with pre-treated wastewater, the biomass yield often increased reaching up to 15 t DM$\cdot$ha$^{-1}\cdot$y$^{-1}$ [21,22]. In the present study, without considering the edge effect of scaling up, the biomass yield was as high as 28 and 34 t DM$\cdot$ha$^{-1}\cdot$y$^{-1}$ for bed A with external winter storage and bed B with internal winter storage, respectively. This might have been the result of high water availability, as the same bed at a lower loading rate and during the same period of operation exhibited a biomass yield of 13 t DM$\cdot$ha$^{-1}\cdot$a$^{-1}$ [1]. However, as in the present study, a high biomass yield was reported by [23] for a short rotation intensive forestry coppice systems in the U.S.

### 4.2. Design Recommendations for Mongolia

Based on the experimental results of the present study and the earlier studies presented in Khurelbaatar et al. [1], the following three major options for up-scaling the wastewater irrigated short rotation coppice systems are recommended to serve as a potential wastewater treatment and biomass production solution for a Mongolian village. Considerations for scaling up must be taken into account, since the recommendation is solely based on the results of a pilot project and the following numbers are extrapolated without considering the edge effects. According to Dolgorsuren et al. [24], there are 64 communities in Mongolia where the treated wastewater is applied to infiltration basins. These treatment plants are, however, either under critical conditions or not in operation. The short rotation coppice systems with wastewater irrigation would be an ideal and "easy-to-build" improvement of the existing situation in terms of both environmental and economical requirements.

As an example, a town of 1500 PE (close to Darkhan city) was chosen to demonstrate the three potential designs of a short rotation coppice system for wastewater treatment and biomass production. MoMo-I [25] reported an average drinking water consumption of 265 L$\cdot$PE$^{-1}\cdot$day$^{-1}$ in Darkhan city. For calculating the wastewater generation of the village, the amount of water consumption per person is multiplied by a factor of 0.825 [26], which resulted in 218.6 L$\cdot$PE$^{-1}\cdot$day$^{-1}$. Assuming that the WWTP should serve 1500 inhabitants, around 120,000 m$^3\cdot$y$^{-1}$ of wastewater should be applied to the system annually. The annual load was calculated based on the estimated hydraulic load and the concentration of primary treated wastewater (see Chapter 2.2).

The three main options presented in Figure 7 are based on the mass removal rates of bed A with external winter storage and bed B with internal winter storage. Another option C with internal winter storage and lower loading rate was included, which was based on a prior study [1], For option B, no external winter storage is required, assuming that the treatment area will serve as an internal winter storage, whereas the design option A includes external winter storage. The dimensions of the winter storage beds were calculated, estimating that 2/3 of the annual wastewater amount had to be stored. This resulted in a storage area of 3 ha with a depth of 2.7 m.

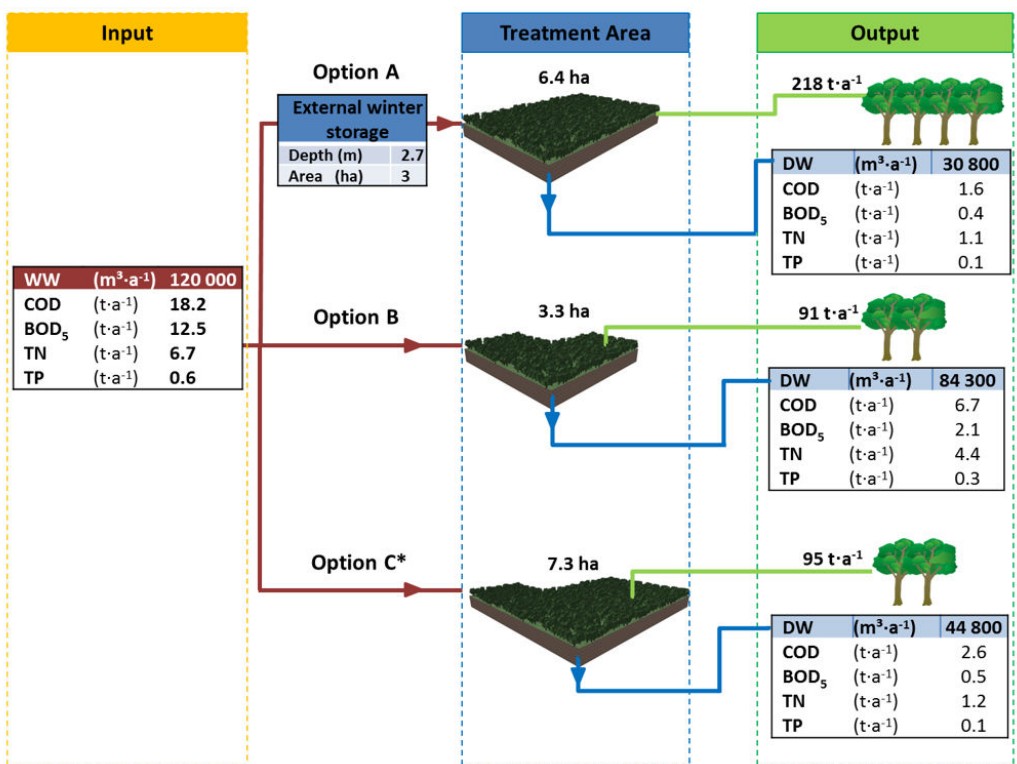

**Figure 7.** Design recommendation for three potential options. The design recommendation for option C* is based on the results reported by Khurelbaatar et al. [1]. Option A represents the external winter storage and only summer irrigation of the short rotation coppice system. Option B and C show the internal winter storage, where the wastewater is stored on the tree-planted area of the short rotation coppice system. The daily loading rate for option A and B are the same, while option C has a lower loading rate.

For each option, the area of the treatment fields was calculated based on the respective hydraulic loading rate (of 15, 15 and 5 mm·day$^{-1}$ for options A, B and C, respectively. Options A and C have similar treatment areas, while option B requires a much smaller area. In spite of the different treatment areas, design options C and B offer similar yields of biomass, whereas option A offers an around two-fold biomass yield. The amount of pollutants associated with drainage water is lowest for option A and highest for option B. Figure 7 presents the mass loadings for primary treated wastewater, the required surface area for the treatment beds, the potential mass of pollutants in drainage water, and the potential biomass yields for each design option.

The irrigation system is not intensively studied in this paper. Most of the irrigation systems used in land application and short rotation forestry systems in other countries are not applicable to the Mongolian situation, which refers in particular to option B and C. The most common and robust irrigation would be a furrow (trench) irrigation system. In the case of option A, pipe and/or drip irrigation can be used. Attention must however be paid to the necessity of emptying the irrigation system before winter.

In addition, it has been observed that the irrigation period for option A can be extended up to 6 months starting in mid-April and ending in mid-October, resulting in smaller winter storage, making this option even more attractive in terms of investment cost.

Option A is recommended for town and municipalities seeking the highest amount of biomass yield accompanied by the lowest risk for groundwater contamination and high investment for the external winter storage. Option B is recommended for municipalities, which have limited space. In spite of the lower reduction rates for pollutants compared to the other two options, it offers a high yield of biomass for a small area. In order to avoid

groundwater contamination, when applying this option, special attention should be paid to the nutrients. The recommended option C is suitable for municipalities with enough available space. By far, it is also the option to reduce the pollutants to a high degree and recycle the wastewater for biomass production.

## 5. Conclusions

Irrigating the short rotation coppice system with primary treated wastewater during summertime only and storing the wastewater in an external winter storage (pond) (option A) might offer a high mass removal rate (i.e., a high removal percentage; reaching up to 91%, 97%, 84%, and 83% for COD, $BOD_5$, TN, and TP, respectively) accompanied by a high biomass yield. In terms of land requirement, this option has however a high demand for area, as it needs additional storage for wastewater during wintertime, which could increase the cost for the system. A low cost alternative would be the irrigation of the short rotation coppice system throughout the year by storing the wastewater on top of the tree-planted area as ice during wintertime. This option B might not only offer a high mass removal rate of the pollutants and low removal percentage (reaching up to 68%, 88%, 37%, and 52% for COD, $BOD_5$, TN, and TP, respectively), but also a high biomass yield within a minimum required area, hence a lower cost solution. In any case, the study has shown that short rotation coppice irrigated with pre-treated wastewater can be operated as a wastewater treatment technology allowing high pollutant removal rates to be achieved, even under the extreme Mongolian climatic conditions. In addition, the tree growth and biomass yield data indicated that both options would offer a very high biomass yield, pointing to the beneficial effects of pre-treated wastewater irrigation on tree production.

**Author Contributions:** M.v.A. and C.F. contributed conception and design of this study. C.M.S., J.A. and G.K. established the pilot study. G.K. performed field and lab analysis as well as the data analysis and wrote the first draft of the manuscript together with C.M.S. and M.v.A., R.A.M. and J.L. provided scientific background and read and commented the manuscript. All authors have read and agreed to the published version of the manuscript.

**Funding:** This study has been completed as part of the project "Integrated Water Resources Management in Central Asia, Model Region Mongolia" MoMo II (FZ: 033L003A), which was financed by the German Federal Ministry of Education and Research (BMBF).

**Institutional Review Board Statement:** Not applicable.

**Informed Consent Statement:** Not applicable.

**Data Availability Statement:** The raw data supporting the conclusions of this article can be made available by the authors upon request. Such enquiries should be directed to the corresponding authors G.K., ganbaatar.khurelbaatar@ufz.de.

**Acknowledgments:** The author would like to thank to German Academic Exchange Service (DAAD) for providing a scholarship and academic support.

**Conflicts of Interest:** The authors declare no conflict of interest. The funders had no role in the design of the study; in the collection, analyses, or interpretation of data; in the writing of the manuscript, or in the decision to publish the results.

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
