# Peer review of "Wastewater Treatment and Wood Production of Willow System in Cold Climate"

_water, doi:10.3390/w13121630_

Round 1

Reviewer 1 Report

In Material and Methods section, authors should specify where the wastewater was coming from.

What was the surfac area of the planted part?

How was the astewater added to the planted parts? Was it disributed on top or to the bottom?

Authors should specify what species of tress were used, just willow and poplar is too vague.

Line 165: drainage water - is the outflow? If so, authors should change the word drainage for outflow

Table 2: The units should indicate that the loads are per year

Lines 301-304: What was the area from which the tress were harvested. Authors should also provide the vaues per m2.

Figure 6: Is the scaling realistic? Would the conditions be the same in a small system as well as in a system with the area of several hectares? How about water distribution?

The manuscript brings interesting results from a region with a specific climatic conditions. In my opinion, the manuscript can be published after a moderate revision.

Author Response

Dear Reviewer,

Thank you for the constructive questions and remarks.

We corrected the manuscript according to your comments and specified the changes with the corresponding locations.

Following you will find the answers and comments to your questions:

Q1: In Material and Methods section, authors should specify where the wastewater was coming from.

A1:The materials and methods section was elaborated with additional text and a graph showing the schematic as well as the cross section of the treatment beds.

Q2: What was the surface area of the planted part?

A2:The surface area of the tree planted beds was 15.75 m² and this is now elaborated in the Material and Methods section, 2.2. Pilot Plant.

Q3:How was the astewater added to the planted parts? Was it disributed on top or to the bottom?

See Q1.

Q4: Authors should specify what species of tress were used, just willow and poplar is too vague.

A4: The Willow and poplar trees have been specified in Materials and Methods.

Q5: Line 165: drainage water - is the outflow? If so, authors should change the word drainage for outflow

A5: Drainage water refers to outflow in this case. However, due to the fact that such systems are not sealed and therefore does not collect the outflow, the term “drainage” is usually the preferred term for this type of systems. Therefore we would like to keep the word drainage.

Q6: Table 2: The units should indicate that the loads are per year

A6: The units are updated as follows: Water in mm/(m²*y), Nitrogen in g/(m²*y)

Q7: Lines 301-304: What was the area from which the tress were harvested. Authors should also provide the vaues per m2.

A7: Thank you for pointing out this important aspect. The values per m² was added.

Q8: Figure 6: Is the scaling realistic? Would the conditions be the same in a small system as well as in a system with the area of several hectares? How about water distribution?

A8: You are of course right bringing a very important aspect here. We pointed out the edge effect were not considered (Line 375-378). We also pointed out the biomass yield were potential number, that might be reached (Line 406-408) in order to avoid giving the readers a “false promise”. The water distribution is also not studied in this paper and must be investigated for scale up systems. This we mentioned in the discussion (line 409).

Reviewer 2 Report

General Comments

The paper deals with deals with a very interesting investigation of the influence of different load conditions for drainage water for low climate temperature. The field of application is very interesting, but the paper requires minor revisions along the following comments before final acceptance:

  1. Please rename Section 2.2 “The pilot plant” as “Pilot plant”.
  2. In section 2.2, a more detailed description of the treatment system should be provided. It should include a schematic representation of the experimental setup (dimensions of treatment units, geometrical configurations of connection pipes, etc. etc.)
  3. The description provided from line 136 to line 180 should be reported in a separate sub-section “Analytical methods”.

Specific Comments

  1. Introduction
  • (Line 90) “This approach is similar to the Zero Discharge Systems reported by …”. By who?
  • (Line 93) Figure number is missing.

  1. Materials and Methods
    • Site description
  • (Lines 117-118) “The climate and precipitation data throughout two years are described in …”. Where? Specify in the text who is being referred to.
    • The pilot plant
  • (Line 156) Insert references on the table for the analytical methods. The equipment should be reported through the text of the manuscript, highlighting what each individual tool was used for.
  • (Lines 162-163) “The detailed procedures of water and biomass analysis are reported by…”. By who?
  • (Line 165) It would be better to edit the formula as an equation, reporting also “Eq. xx” on the side. Furthermore, in the water losses formula, has infiltration in groundwater been included in the drainage water? What about run-off?
  • Line (178) The description of the “analysis campaign” is very poor.

  1. Results
    • Water balance
  • (Lines 196-198) How was the infiltration been measured? This should be better explained in the “Materials and Methods” section.
  • (Lines 204-206) The high pick of infiltration was detected for the option (a)? From graphs, it would seem that the peak in October is more pronounced for option (b). The results in this section should be carefully reviewed and changed when incorrect information is reported…
  • (Lines 206-208) How was the evapotranspiration been measured? This should be explained in the “Materials and Methods” section.

  • Mass removal rate
    • COD & BOD5
  • Change the title of this sub-section “COD & BOD5” to “COD and BOD5
  • (Lines 230-231) Unit of COD removal rate is reported as “g·m-2a-1”. Does "a" refer to the annual period? If it is the case, use “y or year”. The same comment applies to all the other parts of the manuscript where this unit is reported.
  • Lines (231-232) Would it be more correct to consider removed percentage rather than removal efficiency when reporting results? The same comment applies to all the results of the manuscript.
  • Lines (233-234) “The bed operated under option (b) showed a mass removal rate twice higher than that of option (a), …”. The scientific importance of this statement is not clear as the input loads of COD are different ...

  • Nitrogen
  • (Table 2) Add also values of NO3-N

  • coli
  • A part from Table 1, the investigation of this parameter was not previously discussed in the "Materials and Methods" section. So, include also the microbiological analyzes in the discussion.

  1. Discussions
    • Design recommendations for Mongolia
  • (Figure 6) Report in the figure caption the description of cases “a”, “b” and “c”.

  1. Conclusions
  • Some consideration (in terms of observed removal percentages) about the second system (b) should also be included in this section.

Author Response

Dear Reviewer,

Thank you for the constructive questions and remarks.

We corrected the manuscript according to your comments and specified the changes with the corresponding locations.

Following you will find the answers and comments to your questions:

Q1: Please rename Section 2.2 “The pilot plant” as “Pilot plant”.

A1: Corrected

Q2: In section 2.2, a more detailed description of the treatment system should be provided. It should include a schematic representation of the experimental setup (dimensions of treatment units, geometrical configurations of connection pipes, etc. etc.)

A2: An elaborative description including a schematic figure was added to the section.

Q3: The description provided from line 136 to line 180 should be reported in a separate sub-section “Analytical methods”.

A3: It has changed according to the comment.

Specific Comments

Introduction

Q4: (Line 90) “This approach is similar to the Zero Discharge Systems reported by …”. By who?

A4: This and several similar reference mistakes were corrected.

Q5: (Line 93) Figure number is missing.

A5: Figure number was added.

Materials and Methods

Site description

Q6: (Lines 117-118) “The climate and precipitation data throughout two years are described in …”. Where? Specify in the text who is being referred to.

A6: This and several similar reference mistakes were corrected.

The pilot plant

Q7: (Line 156) Insert references on the table for the analytical methods. The equipment should be reported through the text of the manuscript, highlighting what each individual tool was used for.

A7: The analytical method was elaborated, E.coli analytical method was explained.

Q8: (Lines 162-163) “The detailed procedures of water and biomass analysis are reported by…”. By who?

A8: See Q4

Q9: (Line 165) It would be better to edit the formula as an equation, reporting also “Eq. xx” on the side. Furthermore, in the water losses formula, has infiltration in groundwater been included in the drainage water? What about run-off?

A9: The formula was added as an equation and water loss and the runoff was explained.

Q10: Line (178) The description of the “analysis campaign” is very poor.

A10: This section was elaborated and a more humble word was used instead Analysis campaign.

 Results

Water balance

Q11: (Lines 196-198) How was the infiltration been measured? This should be better explained in the “Materials and Methods” section.

A11: The word "Infiltration" was corrected as drainage

Q12: (Lines 204-206) The high pick of infiltration was detected for the option (a)? From graphs, it would seem that the peak in October is more pronounced for option (b). The results in this section should be carefully reviewed and changed when incorrect information is reported…

A12: Thank you for pointing this important mistake out. Now the section was corrected.

Q13: (Lines 206-208) How was the evapotranspiration been measured? This should be explained in the “Materials and Methods” section.

 A13: The Materials and methods was corrected. See explanation of the Equation.

Mass removal rate

COD & BOD5

Q14: Change the title of this sub-section “COD & BOD5” to “COD and BOD5

A14: Corrected

Q15:(Lines 230-231) Unit of COD removal rate is reported as “g·m-2a-1”. Does "a" refer to the annual period? If it is the case, use “y or year”. The same comment applies to all the other parts of the manuscript where this unit is reported.

A15: The unit was corrected for the entire manuscript. Thank you very much pointing this out.

Q16: Lines (231-232) Would it be more correct to consider removed percentage rather than removal efficiency when reporting results? The same comment applies to all the results of the manuscript.

A16: Corrected according to the suggestion, throughout the manuscript.

Q17: Lines (233-234) “The bed operated under option (b) showed a mass removal rate twice higher than that of option (a), …”. The scientific importance of this statement is not clear as the input loads of COD are different ...

 A17: Corrected according to the suggestion.

Nitrogen

Q18: (Table 2) Add also values of NO3-N

A18: NO3-N is part of TN and so is Nh4-n. However the difficulties of presenting NO3-n data is that it is being produced rather than being removed in the effluent. Maybe to be consistent with the data we can remove Nh4-N and present Nitrogen as TN?

E. coli

Q19: A part from Table 1, the investigation of this parameter was not previously discussed in the "Materials and Methods" section. So, include also the microbiological analyzes in the discussion.

A19: An elaboration was added to materials and methods on E.coli.

 Discussions

Design recommendations for Mongolia

Q20: (Figure 6) Report in the figure caption the description of cases “a”, “b” and “c”.

A20: Caption extended with more explanation.

Conclusions

Q21: Some consideration (in terms of observed removal percentages) about the second system (b) should also be included in this section.

A21: The removal percentage was added for option B.

Round 2

Reviewer 1 Report

The authors have done very well necessary revisions, additions. In my opinion, the manuscript can be considered for publication.